# Investigation and Analysis of Wettability, Anisotropy, and Adhesion in Bionic Upper and Lower Surfaces Inspired by Indocalamus Leaves

**DOI:** 10.3390/molecules29153449

**Published:** 2024-07-23

**Authors:** Bo Wang, Donghui Chen, Xiao Yang, Ming Li

**Affiliations:** 1Key Laboratory of Bionic Engineering, Ministry of Education, Jilin University, Changchun 130022, China; wanborn@163.com (B.W.); yangxiao22@jlu.edu.cn (X.Y.); 2College of Biological and Agricultural Engineering, Jilin University, Changchun 130022, China; 3School of Mechanical and Aerospace Engineering, Jilin University, Changchun 130022, China; lmingdr@163.com

**Keywords:** laser scanning, wettability, anisotropy, adhesion, indocalamus leaf

## Abstract

Nature provides us with a wealth of inspiration for the design of bionic functional surfaces. Numerous types of plant leaves with exceptional wettability, anisotropy, and adhesion are extensively employed in many engineering applications. Inspired by the wettability, anisotropy, and adhesion of indocalamus leaves, bionic upper and lower surfaces (BUSs and BLSs) of the indocalamus leaf were successfully prepared using a facile approach combining laser scanning and chemical modification. The results demonstrated the BUSs and BLSs obtained similar structural features to the upper and lower surfaces of the indocalamus leaf and exhibited enhanced and more-controllable wettability, anisotropy, and adhesion. More importantly, we conducted a detailed comparative analysis of the wettability, anisotropy, and adhesion between BUSs and BLSs. Finally, BUSs and BLSs were also explored for the corresponding potential applications, including self-cleaning, liquid manipulation, and fog collection, thereby broadening their practical utility. We believe that this study can contribute to the enrichment of the research on novel biological models and provide significant insights into the development of multifunctional bionic surfaces.

## 1. Introduction

The natural world offers numerous impressive biological models that possess extreme wettability and adhesion properties [1]. For instance, lotus leaves possess excellent superhydrophobic water-repellent and low-adhesion properties for the combination of hydrophobic waxy compounds and multiscale micro/nanostructures on surfaces [2]. Rose petals exhibit superhydrophobicity with high adhesion, enabling water droplets to adhere to the surface without falling off [3,4]. Rice leaves display anisotropic wettability due to their parallelly arranged multiscale micropapillae with nanoprotrusions, resulting in liquid droplets being more likely to slide along the direction parallel to the microgrooves than in the perpendicular direction [5,6]. Previous investigations have found that the excellent wettability, anisotropy, and adhesion observed on the surfaces of typical organisms are attributed to their intricate multiscale micro/nanostructural features and their specialized chemical composition. The synergistic effects of these characteristics enable the realization of diverse functionalities, which are essential for the organisms’ adaptation to their environment and the preservation of survival [7,8].

Many researchers have attempted to mimic the intricate structures of typical creature surfaces that have outstanding self-cleaning [9,10,11], liquid manipulation [12,13], water/fog collection [14,15], and anti-corrosion [16] properties, expanding the potential applications in the automobile, biomedicine, military, and architecture industries [8,17,18]. Ghasemlou et al. reported a method combining soft-imprinting lithography and spin coating techniques to create a lotus-leaf-like superhydrophobic surface with nontoxic and robust mechanical properties [19]. Yun et al. prepared a remarkably stable superhydrophobic surface based on reduced graphene oxide by imitating lotus leaves [20]. Liu et al. proposed a silane-functional polybenzoxazine film with a “petal effect” that possessed good transparent properties and thermal stability [21]. Wu et al. prepared a three-level bionic rice surface and proposed a new method (CADO) by which to characterize, quantitatively, the weak frictional interaction between the droplet and the surface [6]. Lee et al. imitated rice leaves by using electrostatic layer-by-layer assembly on anisotropic microwrinkled substrates and proposed a tunable anisotropic wettability [5]. As for other plants, Long et al. found similar anisotropic behaviors of water droplets sliding down reed leaves and rice leaves and gave a detailed theoretical analysis of such behaviors [22]. Liu et al. proposed a kind of superhydrophobic surface using direct laser interference lithography and hydrothermal treatment inspired from taro leaves, which showed excellent anti-icing properties [23]. However, the current bionic research on plant leaf wettability, anisotropy, and adhesion mostly focuses on lotus leaves, rose petals, rice leaves, etc.; thus, it is of great necessity to investigate more plant leaves with excellent surface functionality. Additionally, a large number of researchers have paid more attention to preparation by imitating one single surface, especially the upper surface of a plant leaf, and there is a lack of studies on the other side (the lower surface). Generally speaking, the two surfaces of one leaf exhibit dissimilar properties to each other, and the reasons for this should be explained. Consequently, systematic research on the wettability, anisotropy, and adhesion of both surfaces of the same leaf simultaneously still needs to be carried out, and the different mechanisms should be investigated in detail. Herein, the indocalamus leaf is proposed as a novel biological model in this study. These leaves are mainly native to the southern region of the Yangtze River in China, including Anhui, Guangdong, and Fujian. Indocalamus leaves have been used in a diverse array of applications, such as beverages, Chinese medicine, and packaging materials. The wettability, anisotropy, and adhesion of their surfaces, however, have been rarely studied, particularly for engineering applications. Thus, the examination of the upper and lower surfaces of the indocalamus leaf holds important research value.

Preparation methods play an important role in achieving the functionality of bionic surfaces. There are many methods for preparing bionic functional surfaces, such as photolithography [6,20,24], template [15,25], electrochemical deposition [26,27], and electrospinning approaches [28,29]. Compared to the methods mentioned above, laser etching emerges as a promising and competitive approach for creating bionic functional surfaces on account of its superior efficiency, wide processing range, low pollution, excellent convenience, and processing accuracy [22,30,31,32,33]. Specifically, the nanosecond laser is capable of facilitating large-scale industrial production due to its lower cost and higher efficiency than picosecond and femtosecond lasers [34,35]. In addition, the laser etching method can not only create typical structures with high accuracy but can also conveniently regulate surface wettability, which has become one of the most widely used processing methods by researchers.

In this work, the indocalamus leaf is innovatively proposed as a biological model used to investigate and imitate the wettability, anisotropy, and adhesion properties of its upper and lower surfaces. Based on the different properties and structural features of the two sides of the indocalamus leaf, bionic upper and lower surfaces (BUSs and BLSs) were prepared via laser scanning and chemical modification. The laser scanning process was illustrated in Figure 1. The as-prepared BUSs and BLSs exhibited similar surface morphologies and enhanced properties to the corresponding upper and lower surfaces of the indocalamus leaf. The performance differences between the BUSs and BLSs were studied through detailed analysis. Contact angles (CAs) and sliding angles (SAs) of the water droplet on BUSs and BLSs in the directions parallel and perpendicular to the microgrooves were systematically measured to analyze the wettability and anisotropy. Dynamic adhesion tests were carried out to investigate surface adhesion. By adjusting the laser scanning interval (SI), surface roughness of the BUSs and BLSs exhibited different characteristics. The results indicated that the BLSs maintained stable wettability, anisotropy, and adhesion due to the larger surface roughness compared to the BUSs, while the BUSs possessed tunable and controllable wettability, anisotropy, and adhesion as the surface roughness decreased. Potential applications of the BUSs and BLSs were verified by self-cleaning, liquid-manipulation, and fog collection experiments. This work can provide more novel ideas and inspirations for the in-depth investigation of bionic functional surfaces which can be widely applied in self-cleaning, water manipulation, and water collection.

## 2. Results and Discussion

### 2.1. Morphologies and Wettability of Upper and Lower Surfaces of the Indocalamus Leaf

Figure 2 displays the morphologies of the upper and lower surfaces of the indocalamus leaf. Microgroove structures on the upper surface were periodically arranged in parallel to the vertical direction. Dry and cracked keratin, formed during the natural drying process of indocalamus leaves, was observed (Figure 2c). Some nanowrinkle structures were visible on the upper surface (Figure 2d). The 3D morphologies of the upper surface and the height of the profile line A are shown in Figure 2i,j, and the microgroove structure was about 270 μm wide and 35 μm deep. Different from the upper surface, the lower surface featured numerous multiscale micro/nanostructures. Microgroove and microcilium structures were arranged in parallel to the vertical direction (Figure 2e,f). The width and depth of the microgroove were about 270 and 43 μm, respectively (Figure 2l). The microcilium structure was about 300 μm long, with an overall upward growth direction (Figure 2f). As shown in the high-magnification SEM images (Figure 2g,h), numerous microprotrusion structures were observed and were mainly composed of microcone and microsphere structures covered with a hydrophobic layer of waxy, dense, nanoscale sheet structures, which were formed by the mixture of large hydrocarbon molecules [15]. The microcone structure was about 25 μm high, and the diameter of the microsphere structure was about 4.5 μm (Figure 2g). Moreover, microgroove, microcilium, and microprotrusion structures were evidently observed in the 3D CLSM image (Figure 2k). The surface roughness of the upper and lower surfaces is listed in Appendix A, and the lower surface possessed much greater roughness than the upper surface.

Figure 3 presents the wettability, anisotropy, and adhesion of the upper and lower surfaces of the indocalamus leaf. The directions parallel and perpendicular to the microgrooves were defined as ‖ and ⊥, indicated by yellow arrows in Figure 2a,e. The CA_⊥_ and CA_‖_ on the upper surface were measured to be 114.8 ± 2.4 and 101.8 ± 1.4°, respectively, indicating hydrophobicity (Figure 3a). Water droplets cannot slide at the tilted angle of 90° in either ‖ or ⊥ directions (Figure 3a). As for the lower surface, the CA_⊥_ and CA_‖_ reached 151.5 ± 1.9° (superhydrophobic) and 138.2 ± 3.2° (nearly superhydrophobic) (Figure 3b). Water droplets slid more easily on the lower surface than on the upper surface. The SA_⊥_ and SA_‖_ were 42.0 ± 1.4° and 24.9 ± 2.2°, respectively (Figure 3b). Moreover, the upper surface exhibited contact anisotropy (CA_Δ_ = CA_⊥_ − CA_‖_ = 13.0°) but no sliding anisotropy (SA_Δ_ = SA_⊥_−SA_‖_ = 0°), while the lower surface possessed both contact anisotropy (CA_Δ_ = CA_⊥_−CA_‖_ = 13.3°) and sliding anisotropy (SA_Δ_ = SA_⊥_−SA_‖_ = 17.1°). The adhesion behaviors of water droplets on the upper and lower surfaces were studied via dynamic adhesion tests (Figure 3c,d). A schematic diagram of the adhesion test and the corresponding testing example are shown in Appendix A. The final states in Figure 3c,d reveal that the residual droplet adhered to the upper surface, whereas it did not adhere to the lower surface and remained a suspended state. Moreover, the length of the droplet stretched on the upper surface was much longer than that on the lower surface in the last contact process. Thus, the upper surface exhibited a larger adhesion force towards the water droplet compared to the lower surface, which can adequately explain the lack of sliding observed in Figure 3a. The above results indicate that the upper surface exhibited hydrophobicity, contact anisotropy, and high adhesion, while the lower surface possessed superhydrophobicity, both contact and sliding anisotropy, and low adhesion. The substantial differences in wettability, anisotropy, and adhesion can primarily be attributed to the disparities in the structural characteristics and roughness between the upper and lower surfaces. The lower surface possessed abundant multiscale micro/nanostructures and greater roughness, which led to a small contact area between the water droplet and the surface. Thus, the lower surface exhibited greater superhydrophobicity and lower adhesion compared to the upper surface. In addition, microgroove-like structures can endow the surface with contact and sliding anisotropy. However, the upper surface did not show sliding anisotropy because the large contact area between the droplet and the upper surface, as well as the lower roughness, prevented the droplet from sliding off in both perpendicular and parallel directions. Based on the different surface morphologies and properties, preparation and investigation on the bionic upper and lower surfaces (BUSs and BLSs) inspired by the indocalamus leaf are carried out in the following sections.

### 2.2. Morphologies of the BUSs and BLSs

As for the Gaussian-mode nanosecond laser, the interactions between materials and the laser can be summarized as two types: gentle ablation and strong ablation [36,37]. When the laser intensity (I) exceeds the strong ablation threshold (I_gth_), a myriad of complex and violent physical processes ensues, including phase explosion, evaporation, melting, ejection, re-solidification, and re-deposition [37,38]. This enables the nanosecond laser to manufacture different kinds of intricate and multiscale structures. Figure 4 and Figure 5 illustrate the SEMs of BUSs and BLSs ablated via laser scanning at SIs of 50, 200, 400, and 800 μm. As shown in Figure 4(a1–d1), it is evident that microgrooves were arranged periodically on each surface. When the pulsed beam was controlled to move in a given direction (line scanning mode in Figure 1), the laser spot with a certain degree of energy interacted with the material surface. A series of complex physical and chemical changes, including phase exploration, rapid evolution, melting, rejection, re-consolidation, and re-deposition, occurred on the area of the laser spot scanning line [37]. Microgroove structures are formed on the original polished surface. Raised ridges formed on both sides of a single microgroove, which was mainly caused by the accumulation of ejected melt and re-solidification during the laser scanning process (Figure 4(b2)). The nanoparticle-like structures densely covered the surface of the ridge structure (Figure 4(a4–d4)). Moreover, some spherical ejected melts solidified on the surface (Figure 4(c2)). As the SI increased, the distance between the areas where the laser spot interacted with the material surface was gradually enlarged, further increasing the distance between the microgrooves. The exposed smooth area between adjacent microgrooves gradually increased (Figure 4(b1–d1)).

Figure 5a–d presents the SEMs of BLSs at SIs of 50, 200, 400, and 800 μm, respectively. A large number of multiscale microprotrusion structures were formed on the whole surface (yellow dashed-line box in Figure 5(b1–d1)). Each microprotrusion structure was encased in a coating of nanoscale particle-like structures (Figure 5(b3–d3), Figure 5(b4–d4)). These micro/nanostructures resulted from the intense interaction between the laser spot and the material, which was caused by the significant overlap of the laser spot during the first scanning [39]. These multiscale microprotrusion structures with nanoscale particle structures were similar to the microcone and microsphere structures covered with a layer of dense, waxy nanoscale sheets on the lower surface of the indocalamus leaf. In addition, microgrooves were also formed on the BLSs (Figure 5(a1–d1)) through the same mechanism as the formation of microgrooves on the BUSs. As for BLS_50_, two raised ridges merged into a ridge between the microgrooves (Figure 5(a2,a3)). The 3D morphologies of the BUSs and BLSs were displayed in Figure 4(a4–d4) and Figure 5(a4–d4), and the corresponding surface roughness is listed in Table 1. When a laser system was used to scan or etch metallic surfaces, a large number of physical and chemical changes occurred on the surface to form micro/nanostructures with different morphologies. The geometric dimensions can be quantitatively estimated using roughness parameters. Generally speaking, the measurement of roughness cannot be fully described by a single parameter; instead, a minimum of three parameters is required [39,40]. In this paper, the Sa (arithmetical mean height), Sz (maximum height), Sq (root mean square height), and Sdr (developed interfacial area ratio) of the BUSs and BLSs were measured. As shown in Table 1, BLSs possessed greater roughness than those of the BUSs processed at the same SI, which showed the same trend as the upper and lower surfaces of indocalamus leaves. Notably, the BUS_50_ and BLS_50_ exhibited only microgrooves, which were different from the typical characteristics observed on the upper and lower surfaces of indocalamus leaves. Consequently, the BUSs (BUS_200_, BUS_400_, and BUS_800_) and BLSs (BLS_200_, BLS_400_, and BLS_800_) were selected to carry out the experiments and analysis on wettability, anisotropy, and adhesion in the following study.

### 2.3. Chemical Compositions of the BUSs and BLSs

EDS and XPS tests were used to characterize the chemical elements on BUSs and BLSs to analyze the change in the surface chemical compositions of both surfaces before and after FAS-17 modification. As shown in Figure 6a,b, EDS results showed that both the BUSs and BLSs possessed five elements before modification, namely, C, Ti, O, V, and Al. After modification, F and Si were found on both BUSs and BLSs. Due to the presence of a large amount of C in FAS-17 molecules, the C content on these two surfaces significantly increased after modification (Table 2 and Figure 6k). The increase in the contents of F, Si, and C led to a decrease in surface O content. Thus, we can preliminarily determine that the FAS-17 molecules assembled on the surfaces. XPS tests were further used to characterize the functional groups on the BUS_200_ and BLS_200_. The corresponding morphologies and surface roughness of both surfaces after modification are shown in Appendix A and Appendix A, respectively. Figure 6c,g shows the survey spectra of BUS_200_ and BLS_200_ before and after modification, respectively. Compared to the peaks of C1s, O1s, and Ti2p on the BUS_200_ and BLS_200_ before modification (the red lines), the peaks of F1s on both surfaces after modification (blue lines) was apparently observed. The high-resolution spectra of C1s of both BUS_200_ and BLS_200_ before modification (Figure 6d,h) can be deconvoluted into three peaks, namely, -C-C-, C-O, and O-C=O [41]. After modification, the C1s included four extra peaks of -C-CF_2_, -CF_2_-CF_3_, -CF_2_, and -CF_3_ (Figure 6e,i), which are characteristic of the FAS-17 molecules [42,43,44,45]. Moreover, F1s proved the existence of C–F, as displayed in Figure 6f,j [41]. All the potential is marked in the figures. Thus, the FAS-17 molecules were successfully immobilized on BUSs and BLSs. The fluorosilanization of BUSs and BLSs is displayed in Figure 6k. First, the triethoxysilane precursor hydrolyzed into active silanol groups. Then, the interfacial condensation and cross-linking reactions took place between the hydroxyl groups on both BUSs and BLSs, resulting in a strong covalent bond formation between the organic layers and the bionic surfaces [45]. A layer of silane molecules forming on the surfaces reduced free energy. It is a well-known fact that the F element can effectively reduce the surface free energy owing to the stable structure of eight electrons with a carbon atom. Other groups can hardly interact with this structure by the van der Waals force. As a result, the chemical modification by FAS-17 can effectively and stably improve surface superhydrophobicity [46]. The stability of the BUS_200_ and BLS_200_ after the abrasion test is shown in Appendix A.

### 2.4. Wettability and Analysis of BUSs and BLSs

Laser-etched surfaces typically exhibit high superhydrophilicity due to the formation of hydrophilic metallic oxides [33]. Chemical modification can effectively realize the transformation from superhydrophilicity (CA < 10°) to superhydrophobicity (CA > 150°) [34,47], which is widely used in the preparation of bionic superhydrophobic functional surfaces [48]. To investigate the wettability of BUSs and BLSs before and after modification, CAs and SAs of water droplets on both surfaces in directions parallel and perpendicular to the microgroove were measured and defined as CA_⊥_, CA_‖_, SA_⊥_, and SA_‖_, respectively. As shown in Figure 7a, BUS_200_ exhibited superhydrophilicity with both CA_⊥_ and CA_‖_ as low as 0° before modification. CA_⊥_ and CA_‖_ on BUS_400_ were 18.6 ± 1.1° and 3.8 ± 0.7°, respectively, while the BUS_800_ exhibited higher angles of 48.9 ± 1.0° and 27.8 ± 1.9° for CA_⊥_ and CA_‖_, respectively (Figure 7b,c). It seemed that the BUSs with lower surface roughness possessed larger CA. However, this phenomenon changed somewhat after modification; both CA_⊥_s and CA_‖_s on BUSs reached superhydrophobicity (CA > 150°), except CA_‖_ on BUS_800_ (129.3 ± 1.9°). Meanwhile, CA_⊥_s and CA_‖_s gradually decreased with the decrease in surface roughness. Distinct from the BUSs, all the BLSs demonstrated uniform CA_⊥_ and CA_‖_ of 0° before modification (Figure 7d–f). The detailed “disappearing process” of the water droplet on BLS_200_ before modification is shown in Figure 7g, with the droplet (4 μL) rapidly spreading around upon contacting the surface within 70ms. After modification, CA_⊥_s and CA_‖_s on the BLSs exceeded 160° and exhibited superhydrophobic stability. This transformation underscored the effectiveness of the chemical modification in endowing the BLSs with superhydrophobicity. Dynamic SAs on BUSs and BLSs after modification are illustrated in Figure 8. SA_⊥_s on BUS_200_, BUS_400_, and BUS_800_ were 10.2 ± 2.0°, 58.3 ± 3.7°, and 90°, respectively, while the SA_‖_s were 6.5 ± 1.3°, 26.8 ± 2.6°, and 90°, respectively (Figure 8a–c). Both the SA_⊥_s and SA_‖_s increased as the surface roughness decreased; in particular, the SA_⊥_ and SA_‖_ of the BUS_800_ exhibited no sliding behaviors, and the water droplet was pinned on the surface. In comparison, the SA_⊥_s on BLS_200_, BLS_400_, and BLS_800_ were 8.2 ± 1.0°, 7.6 ± 1.2°, and 7.1 ± 0.8°, respectively, and the SA_‖_s were 4.7 ± 0.9°, 4.4 ± 0.6°, and 3.6 ± 0.5°, respectively. These values were all less than 10° and lower than those observed on the corresponding BUSs.

According to the above results, the wettability of the BUSs exhibited greater sensitivity and controllability with respect to surface roughness compared to the BLSs, suggesting that the wettability of the BLSs surface is more stable. Different wetting behaviors of BUSs and BLSs were analyzed via the Wenzel and Cassie–Baxter model [49,50]. The wettability of BUSs and BLSs before modification was analyzed via the Wenzel model, as shown in Equation (1):cos*θ*_w_ = *r* cos*θ*,(1)
where *θ*_w_ is the CA of the laser-etched surface, and *r* is the surface roughness factor. *θ* is the equilibrium CA of the polished Ti6Al4V surface, which is hydrophilic (CA = 67.7°) (Appendix A). In accordance with the Wenzel model, surface-wetting behaviors were amplified with the increase in roughness. Specially, as roughness increases, a hydrophilic surface becomes even more hydrophilic, while a hydrophobic surface becomes even more hydrophobic. Consequently, with the decrease in surface roughness from BUS_200_ to BUS_800_ (Table 1), the wettability gradually transformed from superhydrophilicity to hydrophilicity. More importantly, BLS_200_, BLS_400_, and BLS_800_ exhibited sufficiently high roughness to confer upon them superhydrophilic properties with a CA of 0°.

Furthermore, the Cassie–Baxter model was employed to analyze the wettability of the BUSs and BLSs after modification, which suggests that an escalation in surface roughness corresponds to enhanced hydrophobicity, as shown in Equation (2):cos*θ*_CB_ = *f*_s_ (cos*θ*_Y_ + 1) − 1,(2)
where *θ*_CB_ and *θ*_Y_ represent the CAs of the laser-scanned surface and the polished surface after modification; *f*_s_ is the area fraction of solid and liquid surfaces; and *θ*_Y_ is 105.4° (hydrophobic), as shown in Appendix A. According to the CAs on the BUSs and BLSs in Figure 7a–f, the corresponding *f*_s_ values were calculated and are listed in Table 3. For BUSs, the surface roughness decreased with the increase in SI, which further cause the area fraction *f*_s_ to increase significantly. The CA_⊥_s and CA_‖_s of BUSs gradually decreased. Consequently, the BUSs possessed controllable wettability with the decrease in surface roughness and realized the transition from superhydrophobicity to hydrophobicity. As for the BLSs, the relatively smaller contact area fraction (*f*_s_) calculated was caused by the distribution of numerous microprotrusion structures, which possessed sufficiently large surface roughness to support the CA_⊥_s and CA_‖_s being over 160°, stably. In conclusion, the BLSs with micro/nanoprotrusion structures possessed greater roughness, leading to larger CAs in both ⊥ and ‖ directions compared to the BUSs, which successfully imitated the wettability to the upper and lower surfaces of the indocalamus leaf.

### 2.5. Anisotropy and Analysis of BUSs and BLSs

In general, surfaces with microgrooves exhibit anisotropic wettability as the groove-like structures can prevent water droplets from spreading and sliding in the perpendicular direction, allowing for more facile movement in the parallel direction. Consequently, CA_⊥_s and SA_⊥_s are usually larger than the corresponding SA_‖_s and CA_‖_s [5,6,22], which was demonstrated in the last section. Contact anisotropy (CA_Δ_ = CA_⊥_ − CA_‖_) and sliding anisotropy (SA_Δ_ = SA_⊥_−SA_‖_) are commonly employed to quantify the degree of anisotropy in wettability. Figure 8g,h depicts the CA_Δ_s and SA_Δ_s of the BUSs and BLSs. The corresponding schematic diagrams are presented in Figure 8i,j. As for BLSs, both the CA_Δ_s and SA_Δ_s exhibited relatively low values (<10°), and their changing trends were not pronounced from BUS_200_ to BUS_800_. This is attributable to the fact that the width of the microgrooves processed via laser scanning (~58 μm) remained largely unchanged for BLS_200_, BLS_400_, and BLS_800_ (Appendix A and Figure 8j). The CA_Δ_s of the BUSs increased slightly and were followed by a substantial increase, while the SA_Δ_s significantly increased and then decreased to 0° (Figure 8h). The width of the microgroove processed via laser scanning (~32 μm) on the BUSs did not change for BUS_200_, BUS_400_, and BUS_800_; however, the exposed area formed another type of microgroove between adjacent grooves. The widths of the new microgrooves gradually increased and were approximately 162 μm, 372 μm, and 765 μm on BUS_200_, BUS_400_, and BUS_800_, respectively, which were much larger than the microgroove widths on the BLSs. As the water droplets gradually intruded into the microgrooves, the contact and sliding anisotropy markedly increased. Especially, for BUS_800_, the water droplet can fully wet the exposed area and cause very large adhesion, which can cause the droplet to maintain a pinned state and never slide off. The above results fully align with the research of Yong et al. [51] and are similar to the anisotropic wettability of the upper and lower surfaces of indocalamus leaves.

### 2.6. Adhesion and Analysis of BUSs and BLSs

Water adhesion, a critical property of solid materials, has received widespread attention from researchers [52,53]. In the preceding section, the adhesion property could be somehow qualified by the sliding behaviors of water droplets. It was found that the larger SAs were associated with increased adhesion. In order to investigate the differences in adhesion between BUSs and BLSs to the water droplet with greater precision, the adhesion forces between water droplets and bionic surfaces were measured (Figure 9a,b). The adhesion forces of BUS_200_, BUS_400_, and BUS_800_ were 30.2, 78.9, and 115.4 μN, respectively. It can be concluded that the adhesion force significantly increased as the surface roughness decreased for BUSs. On the contrary, adhesion forces of BLS_200_, BLS_400_, and BLS_800_ were 22.9, 18.7, and 17.8 μN, indicating relatively lower adhesion forces. Dynamic adhesion tests were further carried out to illustrate adhesion differences between BUSs and BLSs. In the last contact process (Figure 9d,e), only the droplets were greatly stretched by the large adhesion force on BUS_400_ and BUS_800_. More interestingly, two tiny droplets remained on the surfaces in the final state, while no residual droplets were left on the other four surfaces. Additionally, the residual droplet on BUS_400_ was obviously smaller than that on BUS_800_, meaning the adhesion force on BUS_800_ was larger. The large adhesion force on BUS_800_ can make the droplet tightly adhere to the surface and prevent it from sliding off in both perpendicular and parallel directions on the surface at a tilted angle of 90°, which can explain the lack of sliding anisotropy on BUS_800_ (SA_Δ_ = 0° in Figure 8h). Generally speaking, the presence of large-scale microscale structures enhanced the adhesion of water, as water droplets can penetrate these structures, thereby increasing the contact area between the droplet and the surface [22]. As for BLSs, a large number of rough microprotrusion structures with nanoscale structures were located between adjacent grooves, which can trap abundant air and support the droplet (Figure 8j). The air effect can protect the surface from adhering to the droplet because the air possesses low adhesion to water. As for BUSs, however, the exposed area formed the new large-scale microgrooves and increased the contact area to the water droplet as the surface roughness decreased (Figure 8i). The droplets gradually contacted the exposed area and caused a large adhesion force without the assistance of the stored air. The wider interval led to a stronger adhesion. As a result, the above results revealed that the BUSs with gradually decreasing roughness as SI increased possessed a greater and more controllable adhesion force compared to the BLSs, while the BLSs can always maintain low-adhesion properties.

## 3. Potential Applications of BUSs and BLSs

### 3.1. Liquid-Repellency and Self-Cleaning Property

The superhydrophobic BUS_200_ and BLS_200_ showed good water-repellency properties, as shown in Figure 10a. Various liquid droplets, including deionized water, NaOH (1 mol/L), HCl (1 mol/L), ink, silicon oil (50 cSt), and NaCl (1 mol/L), were capable of maintaining a spherical shape on the BUS_200_ and BLS_200_ compared to the polished surface. Moreover, the self-cleaning test was carried out on BUS_200_, BLS_200_, and the polished surface. Glass powders were used as solid impurities and evenly spread on each surface. The surfaces were held by hands and tilted downwards at a certain angle to the horizontal direction. Water droplets were dropped on the surfaces by a plastic dropper. As shown in Figure 10b–d, once the water droplet contacted the BUSs and BLSs, it immediately took away the glass powders and slid off. An obvious cleaning path appeared on both surfaces during the cleaning process. Finally, both the BUS_200_ and BLS_200_ can be cleaned easily and quickly by water droplets. However, the polished surface cannot be cleaned since the water droplet mixed with the glass powders and adhered to the surface. The solid impurities were not removed, which was caused by the hydrophilicity and high adhesion of the water droplet to the polished surface. The morphology and wettability of the polished surface are shown in Appendix A. The low SAs of BUSs and BLSs were the essential reasons behind the self-cleaning property, which also reflected a very good water-repellency property. The above results indicate that the BUSs and BLSs, with excellent liquid-repellency and self-cleaning properties, can protect the substrate from pollution and, further, be widely used in many engineering fields, such as anti-fouling, self-cleaning, and so on [9,54].

### 3.2. Liquid Droplet Manipulation

The BUSs and BLSs with different degrees of adhesion can potentially be applied in the manipulation of liquid microdroplets and fluids [51]. As shown in Figure 11a, the BLS_200_, BLS_400_, and BLS_800_ with low adhesion force can be used as the workbench for bearing the droplet (5 μL). The BUS_400_ with a large adhesion force (78.9 μN) was used as machine hand A and slowly moved downwards to contact and squeeze the droplet. When the BUS_400_ started to move upwards, the droplet adhered to it completely because of the stronger adhesion force, and it could be transferred from the workbench to machine hand A. Then, the BUS_800_ with the greatest adhesion force (115.4 μN) of all the surfaces was used as machine hand B (Figure 11b). Machine hand A repeated the downward and upward movement, and the droplet adhered to machine hand B. The complete motion process realized the transfer of the droplet from picking to releasing through the two steps described. In addition, the BUSs and BLSs can also be used to merge and mix liquid droplets, which is referred to as the pickup mixing process (Figure 11c). Similar to the previous experiment, a droplet (5 μL) was placed onto the workbench with low adhesion force, and machine hand B, with another water droplet (5 μL), moved downwards. At the moment of contact between the two water droplets, the lower droplet was attracted by the upper droplet and merged into a single larger droplet adhering to machine hand B. The final position of the merged droplet depends on the competing adhesive forces between the workbench and machine hand B [55]. In conclusion, the BUSs and BLSs can realize droplet manipulation, such as droplet pickup, releasing, mixing, and merging, intelligently, which suggests a novel strategy for droplet manipulation.

### 3.3. Fog Collection

A fog collection test was carried out with a homemade water/fog collection test system (Figure 12a). The temperature and relative humidity were 25 °C and 90%, respectively. A commercial humidifier was employed to generate a foggy and humid environment. The flow rate was 175 mL/h. A sample (10 × 10 mm^2^) was fixed vertically by a holder. The outlet of the fog source was in the face of the sample surface at a distance of 10 cm. Beneath the sample was a container for collecting the droplets. The water-collection process was recorded using a CCD industrial microscope. The collected water was measured by a precision balance. Figure 12b presents the results of water collection efficiency on BUSs and BLSs. In general, the fog collection efficiency of the BLSs significantly surpassed that of the BUSs at equivalent SIs. The fog collection efficiency maximized at 934.6 g/m^2^ h on BLS_800_ and minimized at 235.3 g/m^2^ h on BUS_800_. The water collection efficiency of BLS_800_ was about 3.98 times that of BUS_800_. The varying water collection capabilities of BLSs and BUSs can be ascribed to the differences in the adhesion force [56]; the BLSs with hierarchical micro/nanostructures possessed lower adhesion force, which endowed the surface with higher CA and smaller SA, as illustrated clearly above. Actually, the tiny fog droplet that condensed on the BLSs was in a Wenzel state in the initial stage and quickly transitioned to the Cassie state for a more stable condition as the droplet condensation process proceeded [57]. Because the droplets only contacted the top of the micro/nanostructures in a very small contact area, they can easily be removed. Typically, condensed droplets were removed from BLSs during the collection process via two mechanisms: coalescence-induced droplet jumping and gravity-induced droplet sliding (Figure 12c) [58,59]. A lower adhesion force can enhance the two processes, resulting in a more effective water collection ability [59]. In the water collection process on BLSs, the two processes can be easily seen: droplet jumping is marked by yellow-line and dashed-line boxes; and the process of droplet sliding is marked by green-line boxes (Figure 12e–g). A rapid water-removing process can also lead to the surface droplet size of BLSs being much smaller than that of BUSs at the same second and SI, such as BUS_800_ and BLS_800_ at 1200s (comparison to Figure 12g,j). In the case of BUSs, although the plain area between the microgrooves can achieve the quick condensation of tiny droplets [60], droplet removal was more difficult because of the high adhesion force. Larger droplets must be formed via droplet growing and merging to increase the gravity force and overcome the adhesion force (Figure 12d), which greatly shortens the removal time of the droplets while preventing further condensation. Consequently, higher surface adhesion hindered the droplet detachment, thereby decreasing the efficiency of fog harvesting [61]. According to Figure 9a,b and Figure 12b, we can conclude that the water collection efficiency increased as the adhesion force gradually decreased.

## 4. Materials and Methods

### 4.1. Materials

Commercial Ti6Al4V alloy plate (main chemical element including Al of 5.5–6.75%, V of 3.5–4.5%, and Ti of the rest), purchased from Huijing Metal Materials Co., Ltd. (Dongguan, China), was cut into square-shaped sheets (20 mm × 20 mm × 2 mm; 10 mm × 10 mm × 1 mm) via laser cutting. The samples were polished with sandpapers of 1000#, 1500#, and 2000# to eliminate the oxide films and impurities. Polished surfaces were cleaned by an ultrasonic cleaner with acetone, ethanol, and deionized water for 15 min, respectively. 1H, 1H, 2H, 2H-perfluorodecyltrimethoxysilane (FAS-17, 97%) was acquired from Shanghai Macklin Biochemical Technology Co., Ltd. (Shanghai, China). The acetone, ethanol, and deionized water were purchased from Changchun Hongda Fine Chemical Co., Ltd. (Changchun, China). All the reagents were analytical reagents and utilized as received without any additional purification.

### 4.2. Preparation of BUSs and BLSs

A nanosecond laser (FB20-SBGZ, Changchun New Industries Optoelectronics Tech. Co., Ltd., Changchun, China) with a repetition frequency of 20 kHz, a pulse width of 100 ns, a laser beam wavelength of 1064 nm, and a laser spot diameter of 50 μm was used to prepare the BUSs and BLSs. The line scanning of the laser beam was set as the scanning mode. Due to the different surface properties between the upper and lower surfaces of the indocalamus leaf, the preparation processes of BUSs and BLSs were different. As shown in Figure 1, the polished surface was directly processed via one-step laser scanning to fabricate the BUSs, and the laser scanning intervals (SI) were 50, 200, 400, and 800 μm, respectively. The BLSs were fabricated via two-step laser scanning; the polished surface was first scanned at an SI of 3 μm on the surface; then, the as-processed surface was re-scanned at SIs of 50, 200, 400, and 800 μm, respectively. Herein, we define the BUSs fabricated at SIs of 50, 200, 400, and 800 μm as BUS_50_, BUS_200_, BUS_400_, and BUS_800_; similarly, the BLS_50_, BLS_200_, BLS_400_, and BLS_800_ were obtained. Detailed laser processing parameters are listed in Table 4.

### 4.3. Chemical Modification of BUSs and BLSs

In order to prepare the chemical modification solution, 0.5 g of FAS-17 was mixed with 50 g of ethanol in a mass ratio of 1:100 and then subjected to magnetic stirring at room temperature for 30 min. After laser scanning, the as-prepared sample surfaces were immersed into the chemical modification solution for 3 h and then heated in the drying oven at 100 °C for 1 h. Finally, chemically modified BUSs and BLSs were obtained.

### 4.4. Characterization

Surface morphologies were characterized via scanning electron microscopy (SEM, ZEISS EVO 18, Oberkochen, Germany) and field-emission scanning electron microscopy (FESEM, ZEISS MELIN Compact, Oberkochen, Germany). Surface 3D morphologies were observed using a confocal laser scanning microscope (CLSM, Olympus OLS5100, Tokyo, Japan). The chemical composition on the bionic surface was analyzed via an energy dispersive spectrometer (EDS, JEOL JED-2300, Tokyo, Japan) and X-ray photoelectron spectroscopy (XPS, Thermo Fisher Scientific Escalab 250Xi, Waltham, MA, USA). The adhesion force of the BUSs and BLSs was measured by the surface analyzer (LSA100, LAUDA Scientific, Weinheim, Germany). Surface wettability was analyzed by an optical contact angle measurement system (OCAMS, KRüSS DSA 25S, Hamburg, Germany). Five measurements were conducted on the surfaces to reduce the errors. Measurement temperature and relative humidity were 20 °C and 30%, respectively.

## 5. Conclusions

In conclusion, a novel biological model, the indocalamus leaf, was introduced into the bionic preparation of its upper and lower surfaces (BUSs and BLSs) through laser scanning and chemical modification. The wettability, anisotropy, and adhesion properties of BUSs and BLSs were systematically discussed. The following conclusions have been drawn. First, the upper surface of the indocalamus leaf exhibited hydrophobicity, contact anisotropy, and high adhesion, while the lower surface possessed superhydrophobicity, contact and sliding anisotropy, and low adhesion; second, BLSs exhibited more exceptional superhydrophobicity (CAs > 160°, SAs < 10°) due to the larger roughness caused by the dense arrangement of multiscale micro/nanostructures on the surface compared to the BUSs; BUSs showed more controllable wettability (from superhydrophobicity to hydrophobicity) as the water–BUS interface area fraction increased with the gradual decrease in surface roughness; third, excellent anisotropy on the BUSs and BLSs can be achieved via the periodic arrangement of microgroove structures. Both the contact and sliding anisotropy of BUSs dramatically changed compared to those of the BLSs due to the gradually increasing width of the microgroove. Additionally, the BLSs constantly maintained low adhesion (~20 μN) because of the higher surface roughness as the SI increased from 200 to 800 μm, while the BUSs can be gradually tuned from low adhesion (30.2 μN) to high adhesion (115.4 μN). Finally, the BUSs and BLSs can be applied in self-cleaning, liquid manipulation, and fog collection. Through imitating nature and going beyond it, the BLSs exhibited superior wettability, anisotropy, and adhesion, while the BUSs showed more controllable wettability, anisotropy, and adhesion compared to the corresponding upper and lower surfaces of the indocalamus leaf. We believe that this study can contribute to the advancement of research on new biological models and offer significant insights into developing bionic surfaces with multiple functions in practical applications.

## Figures and Tables

**Figure 1 molecules-29-03449-f001:**
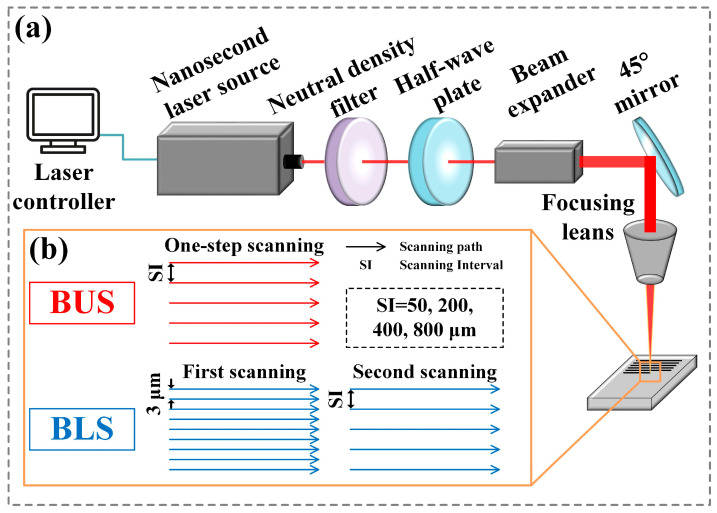
Schematic diagram of the BUSs and BLSs fabricated via laser scanning: (**a**) component units of the nanosecond laser system; (**b**) laser line scanning paths for the preparation of BUSs and BLSs.

**Figure 2 molecules-29-03449-f002:**
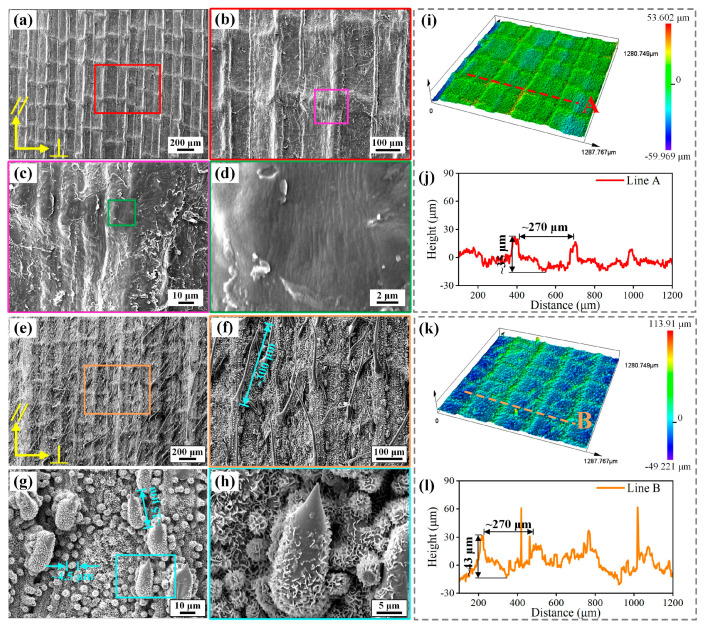
Morphologies of upper and lower surfaces of the indocalamus leaf. (**a**–**d**) SEM images of the upper surface; (**e**–**h**) SEM images of the lower surface; (**i**) 3D CLSM image of the upper surface; (**j**) height of the profile line A corresponding to Figure 2i; (**k**) 3D CLSM image of the lower surface; (**l**) height of the profile line B corresponding to Figure 2k.

**Figure 3 molecules-29-03449-f003:**
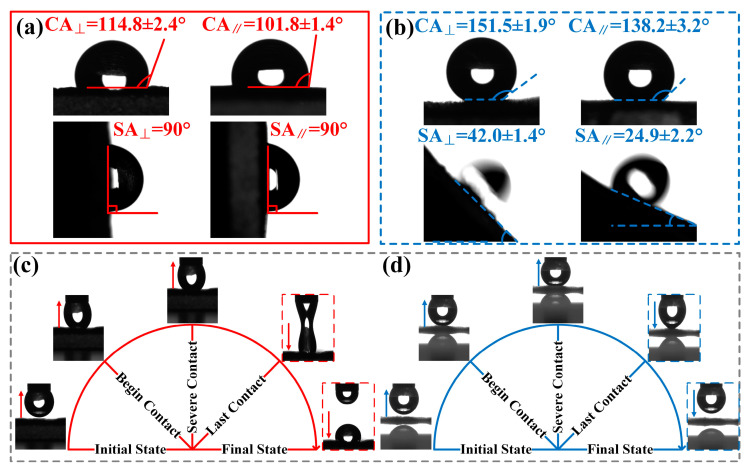
Wettability, anisotropy, and adhesion of the upper and lower surfaces of the indocalamus leaf: (**a**,**b**) CAs and SAs on the upper and lower surfaces; (**c**,**d**) dynamic adhesion behaviors of the upper and lower surfaces.

**Figure 4 molecules-29-03449-f004:**
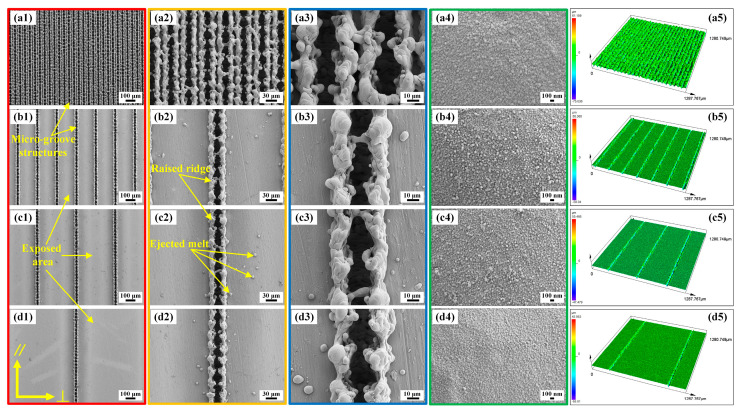
Morphologies of the BUSs: (**a1**–**a4**, **b1**–**b4**, **c1**–**c4**, and **d1**–**d4**) SEM images of BUS_50_, BUS_200_, BUS_400_, and BUS_800_; (**a5**–**d5**) 3D CLSM images of BUS_50_, BUS_200_, BUS_400_, and BUS_800_.

**Figure 5 molecules-29-03449-f005:**
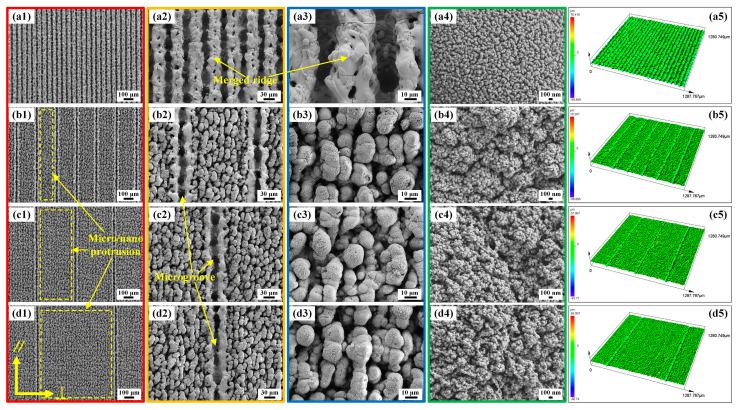
Morphologies of the BLSs: (**a1**−**a4**, **b1**−**b4**, **c1**−**c4**, and **d1**−**d4**) SEM images of BLS_50_, BLS_200_, BLS_400_, and BLS_800_; (**a5**−**d5**) 3D CLSM images of BLS_50_, BLS_200_, BLS_400_, and BLS_800_.

**Figure 6 molecules-29-03449-f006:**
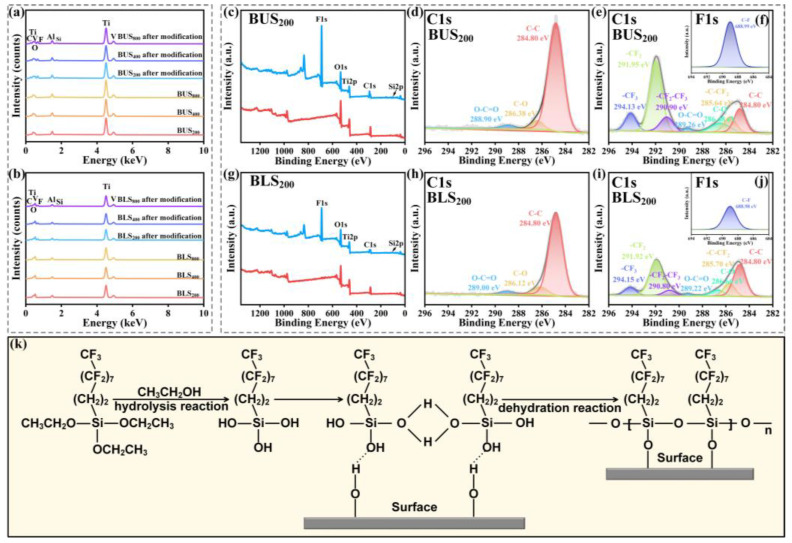
(**a**,**b**) EDS spectra of BUSs and BLSs before and after chemical modification; (**c**–**j**) XPS spectra of the BUSs and BLSs before and after modification; (**k**) mechanism diagram of chemical modification process.

**Figure 7 molecules-29-03449-f007:**
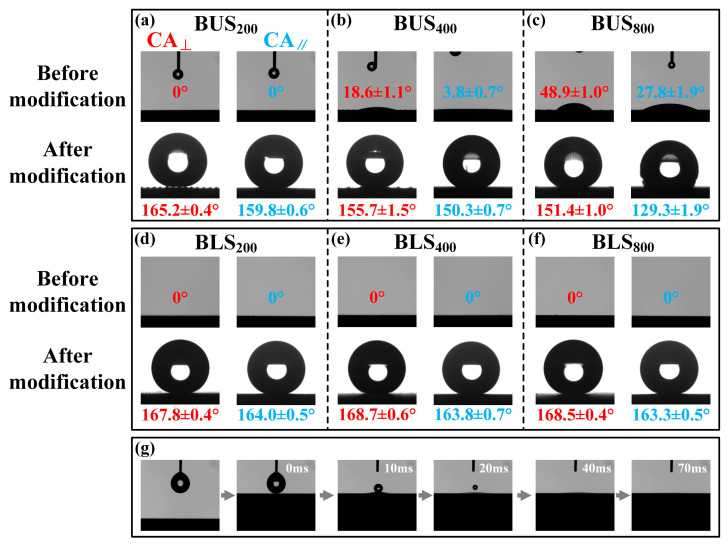
Wettability of the BUSs and BLSs before and after chemical modification: (**a**–**c**) CA_⊥_s and CA_‖_s on the BUSs before and after modification; (**d**–**f**) CA_⊥_s and CA_‖_s on the BLSs before and after modification; (**g**) “Disappearing process” of the water droplet on the BLS_200_ before modification. The volume of the testing water droplet is 4 μL.

**Figure 8 molecules-29-03449-f008:**
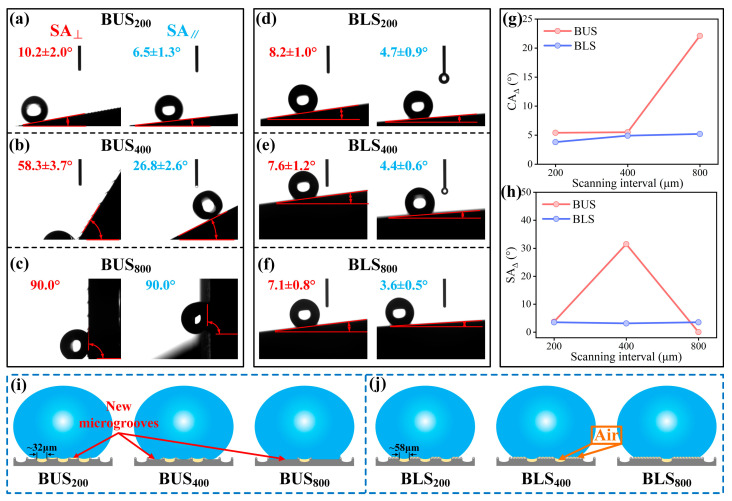
(**a**–**c**) SA_⊥_s and SA_‖_s on the BUSs before and after modification; (**d**–**f**) SA_⊥_s and SA_‖_s on the BLSs before and after modification; (**g,h**) line graph of the CA_Δ_ and SA_Δ_ varying with the SI on BUSs and BLSs; (**i,j**) schematic diagrams of the water droplet contact state with the increase in SI on BUSs and BLSs.

**Figure 9 molecules-29-03449-f009:**
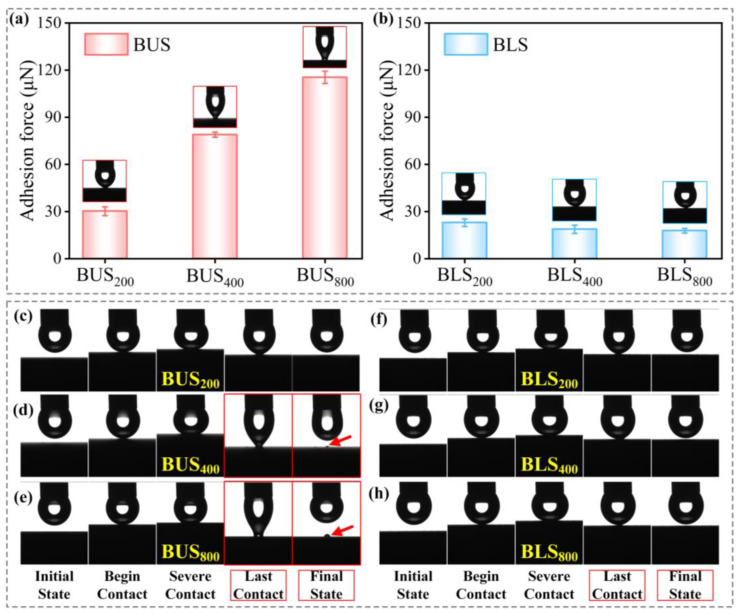
Adhesion property between the water droplet and bionic surfaces: (**a**) bar charts of adhesion force on BUS_200_, BUS_400_, and BUS_800_; (**b**) bar charts of adhesion force on BLS_200_, BLS_400_, and BLS_800_; (**c**–**e**) dynamic adhesion behaviors of BUS_200_, BUS_400_, BUS_800_ (the tiny droplets remaining on the BUS_400_ and BUS_800_ are indicated by red arrows in (**d**,**e**)); (**f**–**h**) dynamic adhesion behaviors of BLS_200_, BLS_400_, and BLS_800_.

**Figure 10 molecules-29-03449-f010:**
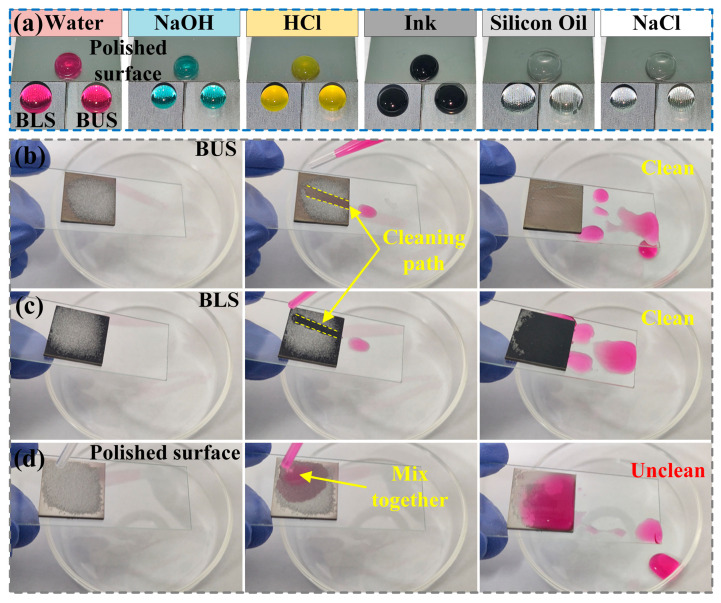
(**a**) Liquid-repellency property of BUS_200_, BLS_200_, and polished surface; (**b**–**d**) self-cleaning property of BUS_200_, BLS_200_, and polished surface.

**Figure 11 molecules-29-03449-f011:**
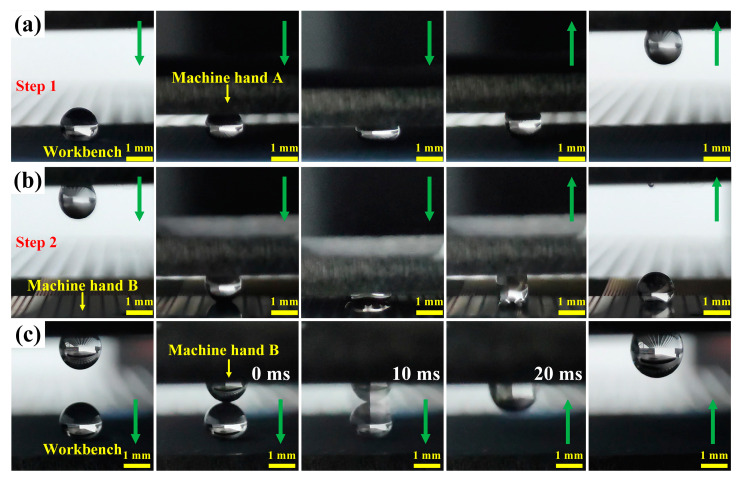
Applications of BUSs and BLSs on droplet manipulation: (**a**,**b**) processes of droplet transference; (**c**) droplet pickup mixing process. The volume of the water droplet is 5 μL, and the green arrows represent the direction of sample movement above.

**Figure 12 molecules-29-03449-f012:**
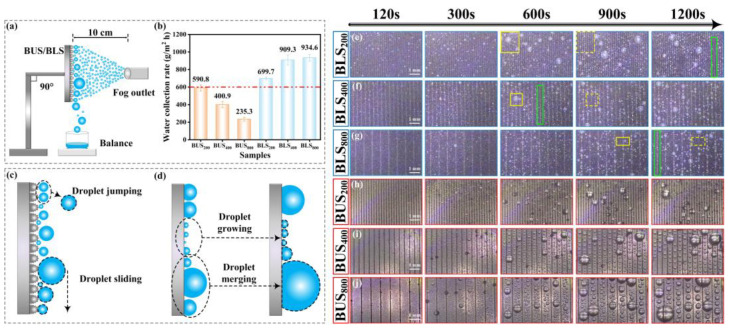
Fog collection test of the BUSs and BLSs: (**a**) schematic diagram of the homemade water/fog collection test system; (**b**) fog collection efficiency of the BUSs and BLSs; (**c**,**d**) processes of the droplet mobility on the BLSs and BUSs; (**e**–**j**) images of fog collection behaviors of BUSs and BLSs from 120 s to 1200 s.

**Table 1 molecules-29-03449-t001:** Surface roughness of BUSs and BLSs with different scanning intervals.

Surface	Sa (μm)	Sz (μm)	Sq (μm)	Sdr (%)
BUS_50_	7.959	123.983	9.966	112.396
BUS_200_	2.383	105.888	4.394	62.631
BUS_400_	1.574	93.548	3.104	51.376
BUS_800_	1.377	92.423	2.672	49.529
BLS_50_	9.223	150.201	11.103	100.988
BLS_200_	5.785	152.086	7.861	85.432
BLS_400_	4.856	137.041	6.543	83.352
BLS_800_	4.889	133.516	6.562	87.281

**Table 2 molecules-29-03449-t002:** Chemical element contents of BUSs and BLSs before and after modification (Mass%).

Bionic Surface/Element	C	O	Ti	Al	V	F	Si
BUS_200_	2.19	12.90	77.03	4.28	3.60	-	-
BUS_400_	2.50	12.72	77.07	4.30	3.41	-	-
BUS_800_	2.76	12.89	76.72	4.26	3.37	-	-
BUS_200_ after modification	4.07	10.54	64.28	3.59	3.01	13.93	0.58
BUS_400_ after modification	3.58	9.99	66.94	3.71	3.15	12.10	0.53
BUS_800_ after modification	3.72	10.38	66.00	3.58	3.04	12.78	0.50
BLS_200_	1.49	29.25	63.00	2.92	3.35	-	-
BLS_400_	1.38	29.38	63.01	2.88	3.34	-	-
BLS_800_	1.68	29.22	62.90	2.91	3.28	-	-
BLS_200_ after modification	3.13	24.76	55.84	2.60	2.77	10.21	0.68
BLS_400_ after modification	2.99	24.98	56.56	2.58	3.01	9.45	0.42
BLS_800_ after modification	3.01	25.00	57.66	2.43	3.05	8.44	0.41

**Table 3 molecules-29-03449-t003:** *f*_s_ values of the BUSs and BLSs calculated via the Cassie–Baxter model.

Surface	BUS_200_	BUS_400_	BUS_800_	BLS_200_	BLS_400_	BLS_800_
CA_⊥_	165.2°	155.7°	151.4°	167.8°	168.7°	168.5°
*f* _s_	4.5%	12.1%	16.6%	3.1%	2.6%	2.7%
CA_‖_	159.8°	150.3°	129.3°	164.0°	163.8°	163.3°
*f* _s_	8.4%	17.9%	49.9%	5.3%	5.4%	5.7%

**Table 4 molecules-29-03449-t004:** Detailed parameters of BUSs and BLSs processed via laser scanning.

Laser Processing Parameter	BUS	BLS	
	One-Step Scanning	First Scanning	Second Scanning
Frequency (kHz)	20	20	
Pulse width (ns)	100	100	
Laser beam wavelength (nm)	1064	1064	
Laser spot diameter (μm)	50	50	
Power (W)	3	3	3
Scanning interval (μm)	50, 200, 400, 800	3	50, 200, 400, 800
Scanning speed (mm/s)	200	200	100
Number of scans	2	2	2

## Data Availability

Data are available upon reasonable request.

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
