# Peer review of "Investigation and Analysis of Wettability, Anisotropy, and Adhesion in Bionic Upper and Lower Surfaces Inspired by Indocalamus Leaves"

_molecules, 2024, doi:10.3390/molecules29153449_

Round 1

Reviewer 1 Report

Comments and Suggestions for Authors

Comments 1: Section 2.2, line 169, “As for a Gaussian-mode nanosecond laser, the interactions between materials and laser can be summarized as two types: the gentle ablation and strong ablation.” Comment: Here, it is necessary to explain what gentle ablation and strong ablation are. Additionally, it needs to be explained which parts of the laser processing are gentle ablation and which parts are strong ablation in this experiment.

Comments 2: Section 2.3, Figure 6, line 254. Comment: SEM and 3D CLSM images of the modified sample by FAS-17 should be added.

Comments 3: Section 3.1, line 409, “Besides, the self-cleaning test was carried out on the BUS200, BLS200, and polished surface.” Comment: The morphologies and wettability (Such as contact angle and slide angle.) of the polished surface should be added to exhibit the difference between BUS and BLS.

Comments 4: Section 3.3, Figure 12, line 493. Comment: There are no diagrammatic notes for figures e to g. And I think the BLS200, BLS200, BLS200, BUS200, BUS200, BUS200 in figures e to g are incorrectly displayed.

Comments 5: Section 4.2, Table 4, line 529. Comment: For the preparation of the BUS and BLS, the authors chose 3W laser power for processing. The effect of other powers on sample performance should be discussed (Such as 1W and 5W).

Comments on the Quality of English Language

Moderate editing of English language required

Reviewer 2 Report

Comments and Suggestions for Authors

1.      In the Introduction, I do not agree with the authors’ statement: “The results showed the BLS maintained stable wettability, anisotropy and adhesion, while the BUS possessed tunable and controllable wettability, anisotropy and adhesion with the increase of laser scanning interval (SI).” Surface properties (wettability, anisotropy and adhesion) are independent of laser scanning interval. The authors confuse the cause-and-effect relationship. The laser scanning interval will change the texture (roughness) and elemental composition. Wettability, anisotropy and adhesion depend on the roughness and elemental composition of the surface. This must be corrected.

2.      In the section 4 Materials and methods, the hydrophobization procedure has not been described. The text does not contain enough information to understand the method of hydrophobization, what is the mechanism of hydrophobization, due to which hydrophobization occurs. Is this a new hydrophobization approach? What is the durability of the resulting hydrophobic surfaces? There is also no information about the errors of the measured parameters, such as roll-ff angle, static angle, etc. The above comments are critical and all questions must be answered in the text of the manuscript.

3.      Figure 2b, the images of roll-of angles are out of focus.

4.      The authors used three-dimensional roughness parameters Sa and Sz. Why were these roughness parameters chosen? They are not enough to fully characterize the roughness. I recommend taking into account the well-known works [D.V. Feoktistov, D.O. Glushkov, G.V. Kuznetsov, D.S. Nikitin, E.G. Orlova, K.K. Paushkina Ignition and combustion characteristics of coal-water-oil slurry placed on modified metal surface at mixed heat transfer // Fuel Processing Technology. Volume 233, August 2022, 107291; Geniy V. Kuznetsov, Dmitry V. Feoktistov, Evgeniya G. Orlova, Kseniya Batishcheva, Sergey S. Ilenok Unification of the textures formed on aluminum after laser treatment // Applied Surface Science. Volume 469, 1 March 2019, Pages 974-982] and add at least Sdr.

5.      Section 2.2. Morphologies of the BUS and BLS. The influence of laser radiation characteristics, including nanosecond duration, on the formation of textures (roughness) has been fairly well studied. As a rule, SEM images are presented in low, medium and high zooming. The authors provided images only in low, medium zoom. By high we mean the field of view of 10 to a maximum of 100 nanometers. There are no such texture images in the manuscript. It is necessary to present them and thereby prove that the texture is multimodal and is capable of demonstrating hydrophobic properties.

But the main complaint is that the authors are trying to find a connection between the characteristics specified by the galvo scanner, which is responsible for the characteristics of the laser beam movement over the surface with the roughness being formed. There can be no such connection. Yes, there are often low-quality articles that allegedly describe such a connection. But the characteristics of laser radiation and the characteristics of the laser beam movement on the surface affect the specific energy in the beam (in the laser spot on the surface). And the formed texture (roughness) depends on this specific energy supplied to the surface, and also depends on the optical and thermophysical characteristics of the surface (surface material). The authors discredit their work by trying to find a connection between the characteristics specified on the galvo scanner with the characteristics of the formed textures. Such a connection cannot exist; it is anti-scientific, anti-physical. It is necessary to rewrite this section taking into account the comment.

6.      Section 2.3. Chemical compositions of the BUS and BLS. The authors make reference 44 and write about the stability of the superhydrophobic coating, but no corresponding experiments have been carried out. It is necessary to either complete the section and conduct research on the persistence of hydrophobic properties, or edit the text of the manuscript.

General opinion about the manuscript. The manuscript has scientific novelty and relevance in the field of studying changes in surface properties by modification with nanosecond laser radiation. But the manuscript requires significant revision based on comments 5 and 6 regarding the cause-and-effect relationship of changes in surface properties. These properties are independent of the characteristics of laser beam movement. Surface properties depend on roughness and elemental composition. Considering these comments, it is necessary to revise the entire text of the manuscript, and not just sections 2.2 and 2.3. This error is contained throughout the entire text of the manuscript.

Comments on the Quality of English Language

Round 2

Reviewer 2 Report

Comments and Suggestions for Authors

I am satisfied with the authors' answers